# Racial and ethnic differences in foveal avascular zone in diabetic and nondiabetic eyes revealed by optical coherence tomography angiography

**Sawarin Laotaweerungsawat**[1,2,3☯], **Catherine Psaras**[1,2☯], **Zeeshan Haq**[1,2], **Xiuyun Liu**[iD][4], **Jay M. Stewart**[iD][1,2]*

1 Department of Ophthalmology, University of California, San Francisco, San Francisco, CA, United States of America, 2 Zuckerberg San Francisco General Hospital and Trauma Center, Department of Ophthalmology, San Francisco, CA, United States of America, 3 Department of Ophthalmology, Charoenkrung Pracharak Hospital, Bangkok, Thailand, 4 Department of Physiological Nursing, University of California, San Francisco, San Francisco, CA, United States of America

☯ These authors contributed equally to this work.
* jay.stewart@ucsf.edu

## Abstract

### Purpose

The purpose of this study was to examine whether racial and ethnic differences in retinal microvasculature are detectable with quantitative measures derived from optical coherence tomography angiography (OCTA).

### Methods

OCTA scans and fundus photography were obtained in 447 eyes from 271 patients with and without diabetes between April and October 2018. Fundus photos were graded by the hospital reading center for diabetic retinopathy (DR) severity. Eight OCTA parameters relating to the foveal avascular zone (FAZ), superficial vascular perfusion, and deep vascular perfusion were analyzed for significant differences between race and ethnicity groups, self-reported by patients and organized according to National Center for Health Statistics groupings. Multiple regression was then used to adjust estimates for possible confounding by age, gender, hypertension, and last hemoglobin A1c level.

### Results

Significant differences in FAZ area were found between white and non-white patients. After adjustment, the differences between white and all non-white groups were statistically significant (p<0.05) among patients with mild to moderate DR. In those without diabetes, the Hispanic and Asian groups had significantly larger FAZ areas (p<0.005) than NH white patients. In those with mild to moderate non-proliferative diabetic retinopathy (NPDR), NH Black, Hispanic, and Asian patients also had significantly larger FAZ areas than NH white patients (p<0.005).

**Data Availability Statement:** All relevant data are within the manuscript and its Supporting Information files.

**Funding:** 1. JMS: That Man May See, Inc. 2. JMS: Research to Prevent Blindness 3. JMS: National Eye Institute, Core Grant for Vision Research EY002162 4. JMS: National Eye Institute, 1R01EY024004 The funders had no role in study design, data collection and analysis, decision to publish, or preparation of the manuscript.

**Competing interests:** The authors have declared that no competing interests exist.

## Conclusion

Significant differences in FAZ area exist among different racial and ethnic groups. These results highlight the importance of considering and further studying race and ethnicity in OCTA analyses of the retinal microvasculature.

## Introduction

Among 20 to 74-year-olds, diabetic retinopathy (DR) is the most common cause of blindness [1]. Hispanic Americans over age 50 have the highest prevalence of diabetic retinopathy in comparison to white Americans, black Americans and Americans of other racial and ethnic groups [2]. The combined prevalence of diabetic retinopathy among all race/ethnicity groups is 5.4%; over age 50, the prevalence is 8% for Hispanic Americans, 5.8% for black Americans, 5.1% for white Americans, and 4.7% for other groups [2]. Previous studies have examined these differences among race and ethnicity groups but have not been able to explain the disparity with traditional diabetic retinopathy risk factors such as age, hemoglobin A1c (HbA1c) levels, and duration of diabetes diagnosis [3].

Prior reports have analyzed race differences in retinal morphology using Spectral Domain Optical Coherence Tomography (SD-OCT) [4–6]. In a study examining the optic nerve and peripapillary region, Poon et. al found that race and ethnicity were associated with differences in the retinal nerve fiber layer (RNFL), neuroretinal rim minimum distance band (MDB) thickness, and area [4]. Knight et. al also found statistically significant differences in the optic nerve head and RNFL thickness measurements between Chinese patients, Hispanic patients, and patients of European and African descent after adjusting for age [5]. Chun et. al in comparing retinal vasculature found that those patients who self-identified as Black versus those who identified as non-Hispanic white had decreased capillary vasculature [7]. However, none of these studies assessed the anatomy of the retinal microvasculature in Hispanic and Asian populations.

Optical coherence tomography angiography (OCTA) is a fairly new and non-invasive technology that allows for the visualization of microvasculature in the retina [8]. In examining microvascular architecture changes in diabetic retinopathy, OCTA has been validated against the gold standard, fluorescein angiography (FA), and found to produce results consistent with those of FA [9]. Algorithms have been developed to quantify different aspects of the captured OCTA scan, and these parameters have been shown to be significantly correlated with the severity of diabetic retinopathy [10, 11].

Achieving a greater understanding of variability in retinal microvasculature among racial and ethnic groups could provide insight into the different impact of DR upon these groups. The purpose of this study was to use OCTA to determine whether racial and ethnic differences exist in quantitative measures of retinal microvasculature in diabetic retinopathy.

## Materials and methods

OCTA scans and relevant data from medical records were collected for this study from Zuckerberg San Francisco General Hospital and Trauma Center from April to October 2018. Diabetic patients were recruited from the diabetic retinopathy screening program, and nondiabetic patients were recruited from the general optometry clinic when patients underwent comprehensive eye exams. Diabetic retinopathy screening patients were not examined.

Only photos were taken as part of the screening program. Patients undergoing comprehensive eye exams had conditions that included but were not limited to dry eye, presbyopia, and glaucoma suspect status. The study was approved by the Human Research Protection Program (HRPP) at the University of California, San Francisco (UCSF). The UCSF HRPP granted a waiver of consent, affirming that patient welfare would not be adversely affected by waiving informed consent. All research adhered to the tenets of the Declaration of Helsinki.

Exclusion criteria for participants without diabetes included any history of ocular injury or ocular disease that could affect the retinal microvasculature such as retinal vascular occlusion, glaucoma, or vitreomacular disease.

Ultra-widefield fundus photography (Optos Daytona, Optos PLC, Dunfermline, UK) and OCTA were obtained in all qualifying subjects. OCTA imaging was performed with a Cirrus$^{TM}$ HD-OCT 5000 with AngioPlex OCT Angiography (Carl Zeiss Meditec, Dublin, CA). Both eyes of each participant were imaged with a scan comprising 245 clusters of B-scans repeated four times, in which each B-scan consisted of 245 A-scans. The resulting OCT volume scan had dimensions of 3 x 3 x 2 mm centered at the fovea. Only images with a signal strength greater than 7, minimal motion artifacts, decentration from the foveal center of less than 20 microns, and minimal evidence of obscuration by media opacities were considered for analysis. Images were taken unmodified for magnification as they were provided by the machine, then cropped the peripheral macula, placed the different eyes in this figure next to each other, locked the aspect ratio in the figure, and then scaled the figure to fit the page.

OCTA parameters were classified into three categories: foveal avascular zone-related metrics, consisting of foveal avascular zone (FAZ) area, FAZ acircularity index (FAZ ACI), and FAZ circularity index (FAZ CI); vessel density-related metrics, consisting of superficial perfusion density (SPD), superficial vessel density (SVD), deep perfusion density (DPD), and deep vessel density (DVD); and a nonperfusion metric, the total extrafoveal avascular area (tEAA). CIRRUS 11.0 software automatically calculated the FAZ area, FAZ CI, perfusion density, and vessel density of the superficial capillary plexus. Processing of these scans and calculations of parameters has been described previously [12].

Age, duration of diabetes, hypertension status, and HbA1c level, when available, were noted for each participant from the patient medical record. Myopia status was calculated from axial length using 25.9 mm for men and 25.3 mm for women [13]. Duration of diabetes and year of diabetes diagnosis were self-reported by the patient and verified in the medical record when possible. Race and ethnicity were self-reported orally during patient intake in the medical record. Race was defined using National Center for Health Statistics (NCHS) grouping, and ethnicity was defined as being of Hispanic or Latino or not [14]. Race/ethnicity as it is defined in this study is a social construct and is not a marker for genetics [15]. Race/ethnicity categories were used in agreement with that used by the Centers for Disease Control (CDC) National Center for Health Statistics (NCHS) [16]. Best-corrected visual acuity (BCVA) was measured for each eye using the Snellen chart and converted into logMAR visual acuity using methods previously described [17]. DR severity was graded using the Scottish Grading Protocol [18] from the color fundus photos by the department's DR screening program reading center, in which patients with diabetes undergo ultra-widefield fundus photography, and then the photographs are graded in standardized fashion by trained and supervised graders. DR severity for each patient was assigned into one of three groups: nondiabetic participants (control), diabetes patients without retinopathy, and mild to moderate non-proliferative diabetic retinopathy (NPDR) [18]. Proliferative diabetic patients were excluded due to the small sample size.

Stata 17.0/BE (StataCorp LLC, College Station, TX, USA) was used to perform the statistical analysis. A p-value of less than 0.05 was considered statistically significant for all statistical tests. Analyses regressing the OCTA parameter on Hispanic, NH-Black, and Asian indicator

variables with NH White as the reference group were used to analyze whether there was a significant difference in the OCTA parameters between at least two of the four patient groups. The Bonferroni correction was utilized prior to assessing statistical significance of linear regression results to account for multiple comparisons. Multivariable regressions, stratified by stage of DR disease, were used to adjust for age, gender, hypertension status, and HbA1c levels. Linear combinations of coefficients were examined post estimation to assess differences between all race/ethnicity groups. For the multivariate regressions, both eyes, if available, were included for each patient. Standard errors in all regressions were adjusted for patients who had two eyes included in the study using the Huber White sandwich estimator [19].

## Results

447 eyes from 271 patients were included in this study. Summary statistics for all included patients are presented in Table 1.

Overall quality of the usable images was good. Less than 5% had a signal strength of 7 or 8, 15% had a signal strength of 9, and approximately 80% had a signal strength of 10. The source data used in the analysis is available online in S1 Table.

A linear regression, which was unadjusted for stage of disease, showed a significant difference in FAZ area among the four race/ethnicity groups (NH white, NH black, Hispanic, and Asian) (Table 2). All other OCTA parameters did not show significant differences among the race groups. For FAZ area, NH white patients had the smallest FAZ area (0.22 mm$^2$ ± 0.09 mm$^2$) while Hispanic patients had the largest FAZ area (0.33 mm$^2$ ± 0.12 mm$^2$).

After stratifying by disease severity, there were noticeable differences among race/ethnicity groups in both those without diabetes and those with mild to moderate NPDR (Table 3). After

**Table 1. Summary of patient characteristics.**

|  | Total | NH White | NH Black | Hispanic | Asian | p-value |
|---|---|---|---|---|---|---|
| **Eyes** | N = 432 | N = 74 | N = 37 | N = 225 | N = 96 | |
| **Patients** | N = 271 | N = 44 | N = 22 | N = 140 | N = 65 | |
| **Age** | 53 (13) | 52 (12) | 50 (11) | 50 (13) | 59 (11) | <0.001 |
| **Gender** | | | | | | 0.011 |
| Male | 200 (46%) | 47 (64%) | 14 (38%) | 99 (44%) | 40 (42%) | |
| Female | 232 (54%) | 27 (36%) | 23 (62%) | 126 (56%) | 56 (58%) | |
| **Hypertension** | 219 (52%) | 44 (65%) | 13 (38%) | 113 (51%) | 49 (52%) | 0.071 |
| **Diabetes Status** | | | | | | <0.001 |
| No Diabetes | 123 (28%) | 38 (51%) | 8 (22%) | 54 (24%) | 23 (24%) | |
| Diabetes (No DR) | 232 (54%) | 22 (30%) | 22 (59%) | 130 (58%) | 58 (60%) | |
| Mild to Moderate DR | 77 (18%) | 14 (19%) | 7 (19%) | 41 (18%) | 15 (16%) | |
| **Years Since Diagnosis** | 8 (8) | 8 (6) | 9 (7) | 7 (9) | 8 (6) | 0.90 |
| Missing, n (%) | 108 (40%) | 12 (27%) | 10 (45%) | 57 (41%) | 29 (45%) | |
| **Last HbA1c*** | 8.1 (2.2) | 8.7 (2.1) | 8.4 (1.8) | 8.3 (2.5) | 7.1 (1.4) | <0.001 |
| **LogMAR**, best eye | 0.05 (0.09) | 0.03 (0.06) | 0.04 (0.06) | 0.06 (0.11) | 0.07 (0.09) | 0.034 |
| **Myopia Status** | | | | | | 0.60 |
| No Myopia | 269 (62%) | 51 (69%) | 22 (59%) | 139 (62%) | 57 (59%) | |
| Myopia | 163 (38%) | 23 (31%) | 15 (41%) | 86 (38%) | 39 (41%) | |

Abbreviations: SD = standard deviation, HbA1c = hemoglobin A1c, VA = visual acuity, DR = diabetic retinopathy, NH = non-Hispanic; Data are presented as mean (SD) for continuous measures, and n (%) for categorical measures; p-values are from ANOVA for continuous variables and Pearson's chi-squared tests for categorical and binary variables

* = HbA1c values only available for diabetic patients; Missing = datapoint missing from patient electronic health record.

**Table 2. Mean values for OCTA parameters by race/ethnicity group.**

| Group | tEAA | SVD | SPD | FAZ area** (mm²) | FAZ CI | FAZ ACI | DVD | DPD |
|---|---|---|---|---|---|---|---|---|
| **NH White** (N = 74) | 0.04± 0.09 | 20.7 ± 1.61 | 37.9 ± 2.49 | **0.22 ± 0.09** | 0.64 ± 0.11 | 1.26 ± 0.15 | 14.9 ± 1.56 | 30.6 ± 3.33 |
| **NH Black** (N = 37) | 0.05± 0.08 | 21.2 ± 1.38 | 38.4 ± 2.15 | **0.32 ± 0.12** | 0.65 ± 0.08 | 1.24 ± 0.08 | 14.9 ± 1.27 | 30.0 ± 2.82 |
| **Hispanic** (N = 225) | 0.04 ± 0.14 | 21.1 ± 1.56 | 38.3 ± 2.36 | **0.33 ± 0.12** | 0.67 ± 0.09 | 1.24 ± 0.12 | 14.6 ± 1.37 | 29.4 ± 2.90 |
| **Asian** (N = 96) | 0.02 ± 0.04 | 20.9 ± 1.60 | 37.9 ± 2.56 | **0.32 ± 0.10** | 0.67 ± 0.07 | 1.23 ± 0.08 | 14.6 ± 1.19 | 29.2 ± 2.49 |

N = number of patients, one eye per patient is included even if scans available on both; Data format: Mean ± SD

** = denotes significant difference among groups (p<0.05) using linear regression. NH: non-Hispanic; tEAA: total extrafoveal avascular area; SVD: superficial vessel density; SPD: superficial perfusion density; FAZ: foveal avascular zone; FAZ CI: FAZ circularity index; FAZ ACI: FAZ acircularity index; DVD: deep vessel density; DPD: deep perfusion density.

adjusting for age, gender, hypertension, and HbA1c these differences persisted in the mild to moderate DR group. Non-white patients in both those without diabetes and those with mild to moderate NPDR had larger FAZ areas than NH white patients (Table 3). Among patients without diabetes, there were no significant differences between Hispanic and NH-Black (mean difference: 0.06 (95% CI: -0.05–0.16), Hispanic and Asian (mean difference: 0.02 (95% CI: -0.07–0.10), and Asian and NH Black patients (mean difference: 0.07 (95% CI: -0.07–0.10). Among, patients with mild to moderate NPDR, there were also no significant differences between Hispanic and NH-Black (mean difference: 0.10 (95% CI: -0.01–0.20), Hispanic and Asian (mean difference: 0.04 (95% CI: -0.04–0.12), and Asian and Hispanic patients (mean difference: 0.10 (95% CI: -0.01–0.20). Among patients with diabetes without retinopathy, the results remained similar (FAZ Area$_{Asian}$- FAZ Area $_{NH Black}$ = 0.003 (95% CI: -0.07–0.06); FAZ Area $_{Hispanic}$- FAZ Area $_{NH Black}$ = 0.01 (95% CI: -0.07–0.05); FAZ Area $_{Asian}$- FAZ Area $_{Hispanic}$ = 0.01 (95% CI: -0.05–0.04)). There results were adjusted for age, gender, hypertension, and HbA1c where appropriate.

**Table 3. Difference in FAZ area (mm²) between race/ethnicity group and NH white group, adjusted and unadjusted.**

| | NH White | | NH Black | | Hispanic | | Asian | |
|---|---|---|---|---|---|---|---|---|
| | Unadj. | Adj | Unadj. | Adj | Unadj. | Adj | Unadj. | Adj |
| No Diabetes | Ref | Ref | 0.06 | 0.05[a] | 0.12** | 0.11[a]** | 0.12** | 0.12[a]** |
| | | | [-0.04–0.16] | [-0.05–0.15] | [0.07–0.17] | [0.06–0.15] | [0.04–0.19] | [0.04–0.20] |
| Eyes | 38 | 36 | 8 | 8 | 54 | 54 | 23 | 23 |
| Diabetes (No Retinopathy) | Ref | Ref | 0.05 | 0.05[b] | 0.06 | 0.06[b]* | 0.05 | 0.06[b] |
| | | | [-0.03–0.13] | [-0.02–0.12] | [-0.01–0.12] | [0.01–0.12] | [-0.02–0.12] | [-0.01–0.11] |
| Eyes | 22 | 18 | 22 | 19 | 130 | 129 | 58 | 57 |
| Mild to Moderate NPDR | Ref | Ref | 0.29** | 0.23[b]** | 0.16** | 0.13[b]** | 0.14** | 0.17[b]** |
| | | | [0.19–0.40] | [0.11–0.35] | [0.07–0.26] | [0.05–0.22] | [0.05–0.22] | [0.09–0.26] |
| Eyes | 14 | 14 | 7 | 7 | 41 | 38 | 15 | 15 |

[a]: adjusted for age, gender, and hypertension

[b]: adjusted for age, gender, hypertension, and HbA1c

* = denotes statistical significance at P < 0.05

** = denotes statistical significance at P < 0.005; SE adjusted for patients with two eyes; HbA1c = hemoglobin A1c; Unadj. = Unadjusted; Adj. = Adjusted; NH = Non-Hispanic.

## Discussion

This study evaluated the differences in retinal vasculature among race groups using OCTA imaging. In our study, the only OCTA parameter affected by race/ethnicity was FAZ area. After adjusting for age, gender, hypertension status, and last HbA1c values, there were significant differences between white and non-white patient groups. An earlier report found that microvasculature in the retina is affected by high myopia [20]. In the present study, however, there was no significant difference in myopia rates among the four race/ethnicity groups. NH white patients had the smallest FAZ area while NH black patients had the largest FAZ area in patients with DR. When stratified by disease severity, these racial and ethnic differences persisted in all disease severity groups.

Previous studies have found significant differences between retinal morphology of white and African American patients [21, 22] and white, African American, and Hispanic patients [23]. Kelty et. al found that white patients had a 32 μm greater mean foveal thickness (MFT) than African American patients (217 ± 25 μm vs. 185 ± 17 μm; P < 0.001). This would be consistent with our finding of a significantly smaller FAZ in white patients, in that a greater mean foveal thickness would correlate with a smaller FAZ. Poon et al., in examining race differences in retinal morphology as it relates to glaucoma, found that the retinal nerve fiber layer (RNFL) scans and the MDB from SD-OCT were significantly affected by race and ethnicity [4]. In comparison to white participants, Asian and Hispanic participants had a greater mean RNFL thickness on a 2D circle scan. Also in comparison to white participants, Black and Asian participants had smaller mean MDB thickness and area on 3D volume scans. These findings are in agreement with the present study. Poon et. al only found one significant difference between Hispanic and White participants. Hispanic participants made up the smallest share of the study population (272 total participants: 64.3% White, 14.7% Black, 14.7% Asian, 6.3% Hispanic) and the Poon et. al study may have been underpowered to detect differences in this group. In the present study, Hispanic participants made up the largest share of the study population. The difference in population makeup between these studies highlights the importance of adequate statistical power for drawing conclusions about meaningful differences between groups. Depending upon the geographic location of a study center and its attendant patient population demographics, certain groups may be relatively over- or under-represented. In this instance, different retinal parameters were analyzed in the report of Poon et. al versus the current study, making it difficult to compare the studies directly, but taken together the studies support the notion that racial and ethnic differences can contribute to reproducible differences in retinal anatomy that could have implications for disease susceptibility or progression.

Diabetic retinopathy alters the anatomy of the fovea and in particular the foveal avascular zone [24, 25]. Previous studies have examined FAZ as a predictor for disease severity [10, 26–28]. FAZ area has been shown to be a more strongly associated with central macular thickness (CMT) and sex than with other FAZ parameters such as the circularity of the FAZ [29]. A previous study has shown positive correlations between CMT and ethnicity, body mass index, smoking, and older age [30]. As pointed out by Dubis et. al, it is therefore important to distinguish the difference between pathological changes in the FAZ and those possibly associated with other patient characteristics [31]. Previous studies have been limited by little racial or ethnic variation in study participants and thus were unable to study contributors associated with race/ethnicity to retinal morphology as captured by OCTA [32–36]. Wylęgała et. al compared the FAZ of Polish Caucasians and Han Chinese patients and found a smaller mean FAZ area among the Caucasian patients [37]. This agrees with the findings of our study. The present study, demonstrating baseline differences in FAZ area between racial and ethnic groups, highlights the importance of considering race/ethnicity when studying the FAZ with OCTA

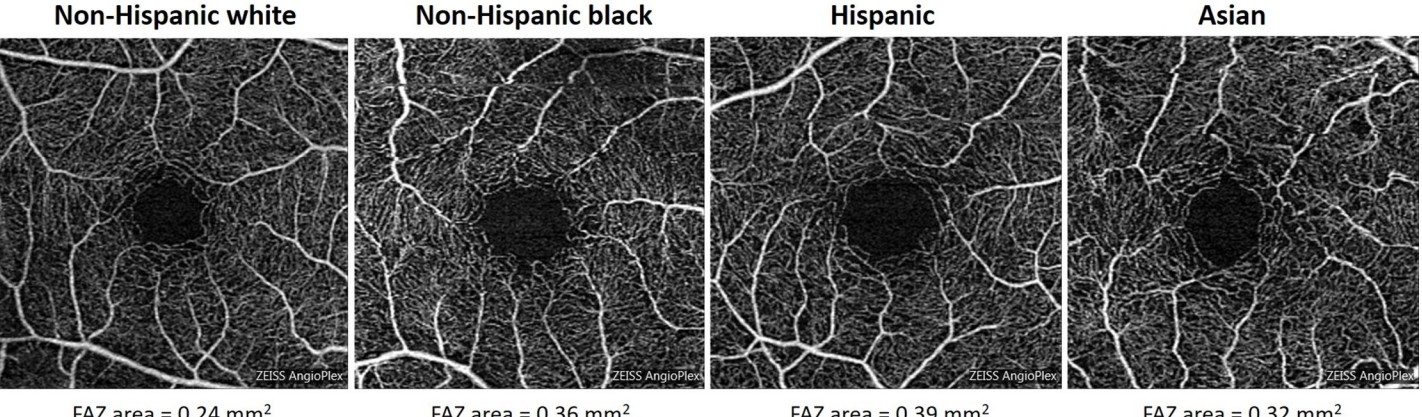

**Non-Hispanic white**  FAZ area = 0.24 mm²

**Non-Hispanic black**  FAZ area = 0.36 mm²

**Hispanic**  FAZ area = 0.39 mm²

**Asian**  FAZ area = 0.32 mm²

**Fig 1. Representative cases demonstrating FAZ differences among ethnic groups in non-diabetic patients.**

angiography. Though race might not be a biological determinant as it has been captured in this study, further research is needed to elucidate more details on the etiology of these findings.

In this study, Hispanic, NH black, and Asian patients had larger FAZ areas at baseline (that is, in non-diabetic subjects), in comparison to NH white participants (Fig 1 and Table 3). Along with these baseline differences among populations, we also found evidence that DR disease can differentially affect the FAZ of white and non-white patients; specifically, the difference between white and non-white patient groups increased once DR disease was present. In future studies, it would be worth exploring whether a larger FAZ at baseline has less reserve or is more predisposed to pathological enlargement with progressive diabetes-related damage.

For each racial and ethnic group, a representative OCTA image was selected. FAZ area is shown for each.

## Limitations

A limitation of this study is that because the race and ethnicity status of subjects was self-reported, we cannot say that these data accurately correlate with biologic and genetic patterns. We cannot separate the effects of possible determinants associated with race/ethnicity with self-reported race due to a lack of granularity on other factors [15, 38]. As noted in Jones (2001), race, as used in this study, may be capturing context of patients' situations instead of patient characteristics such as differential access to ophthalmological and diabetes care [38]. This likely was not fully captured by hypertension status and recent HbA1c values.

There was also an imbalance between race/ethnicity groups. This imbalance is present because this sample was a representative sample from the hospital clinic and not one where patients were recruited to reach equal representation among groups. Thus, there are race/ethnicity groups with relatively few numbers of patients. It is possible that there were groups with too few participants to detect any convincing differences.

In the present study, non-White race/ethnicity groups were similar to one another, but different to the non-Hispanic White group, particularly in the diabetic group. This was consistent with the prevalence of diabetes without retinopathy in the study population but is incongruous with the differences in diabetic retinopathy prevalence in the general population. Further study is needed to explore these findings.

Additionally, patients in the diabetic groups with comorbid conditions affecting the retinal vasculature were not able to be excluded *a priori*, as they were in the non-diabetic control

group, since this information was not recorded as a separate field under the diabetic retinopathy screening protocol. For example, Shokr et. al found that those patients with dry eye disease had greater retinal vascular impairment than controls [39]. However, it is unlikely that any patients with relevant conditions affecting the retinal vasculature were included in the diabetic groups in this study, because such findings would have been noted on the fundus photograph grading report, and there were no instances in which this occurred. However, further study accounting for these factors is warranted.

## Conclusions

This study found that there were differences associated with race/ethnicity in the FAZ of patients with and without diabetic retinopathy. These differences persisted even after adjusting for age, gender, hypertension status, and last HbA1c values among non-diabetic Hispanic, Asian, NH White patients, and among all groups in comparison to NH white patients among those with mild to moderate NPDR. This suggests that race and ethnicity may be important factors to consider for further study when examining foveal morphology in diabetic retinopathy disease pathogenesis as well as more broadly in the study of retinal vascular OCTA imaging.

## Supporting information

**S1 Table. Supporting data.** The source data used in the analysis.
(XLS)

## Author Contributions

**Conceptualization:** Sawarin Laotaweerungsawat, Jay M. Stewart.

**Data curation:** Sawarin Laotaweerungsawat, Catherine Psaras.

**Formal analysis:** Catherine Psaras, Zeeshan Haq, Xiuyun Liu.

**Funding acquisition:** Jay M. Stewart.

**Investigation:** Sawarin Laotaweerungsawat.

**Methodology:** Sawarin Laotaweerungsawat, Catherine Psaras, Zeeshan Haq, Xiuyun Liu, Jay M. Stewart.

**Resources:** Jay M. Stewart.

**Supervision:** Jay M. Stewart.

**Writing – original draft:** Catherine Psaras.

**Writing – review & editing:** Zeeshan Haq, Jay M. Stewart.

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
