## [Decision Letter · Decision Letter 0]

30 Jul 2021

PONE-D-21-20816

Racial and ethnic differences in foveal avascular zone in diabetic and nondiabetic eyes revealed by optical coherence tomography angiography

PLOS ONE

Dear Dr. Stewart,

Thank you for submitting your manuscript to PLOS ONE. After careful consideration, we feel that it has merit but does not fully meet PLOS ONE’s publication criteria as it currently stands. Therefore, we invite you to submit a revised version of the manuscript that addresses the points raised during the review process.

Both reviewers found this an interesting study and made several comments to make it better. We look forward to the revised version 

We look forward to receiving your revised manuscript.

Kind regards,

Demetrios G. Vavvas

Academic Editor

PLOS ONE

Journal Requirements:

“1. JMS: That Man May See, Inc.

2. JMS: Research to Prevent Blindness

3. JMS: National Eye Institute, Core Grant for Vision Research EY002162

4. JMS: National Eye Institute, 1R01EY024004

We note that you received funding from a commercial source: That Man May See, Inc.

Reviewers' comments:

Reviewer's Responses to Questions

**Comments to the Author**

1. Is the manuscript technically sound, and do the data support the conclusions?

Reviewer #1: Yes

Reviewer #2: Partly

2. Has the statistical analysis been performed appropriately and rigorously? 

Reviewer #1: Yes

Reviewer #2: Yes

3. Have the authors made all data underlying the findings in their manuscript fully available?

Reviewer #1: Yes

Reviewer #2: Yes

4. Is the manuscript presented in an intelligible fashion and written in standard English?

Reviewer #1: Yes

Reviewer #2: Yes

5. Review Comments to the Author

Reviewer #1: This is an interesting study with useful findings. Here are my comments and suggestions.

Line 65: add reference or connect the 2 sentences

Lines 80-81: In fact, in the study of Chun et al (PMID: 31596848) differences in macular capillary parameters between healthy black and white subjects were assessed using OCTA.

Lines 103: In the recent study of Shokr et al (PMID: 33576186), it was suggested that dry eye disease is associated with retinal microvasculature dysfunction. Even though the methods for assessing retinal microvasculature are different, it would be interesting to account for this factor as well either to only the nondiabetic group or also to diabetics if data available.

What about myopia? Are they any differences among groups? OCTA studies have shown that there are significant microvascular retinal alterations in highly myopic eyes (e.g. PMID: 27820633).

Was adding only the right eye based on any previous studies or not? You could still randomly select right-left.

Lines 138-140: Please, give more information about the DR screening program reading center.

Lines 140-142: Which grading system did you use? What about proliferative DR?

Lines 142-144: You could move this sentence to the next paragraph where you are mentioning the statistical softwares and the tests that you used.

It seems like that even though you collected data from 280 patients, since you applied one-way ANOVA you ended up analyzing data of almost half of the initial cohort. Is there a specific reason why you used this approach instead of using multilevel models also for the analysis of OCTA parameters which is actually the main purpose of your study?

Additionally, you mention that 280 patients were included but in table 2 you analyzed data of 271 patients (one eye). If so, you will need to update the demographic table in order to reflect the data presented in table 1 and also either change the number of patients included or explain what happened to those 9 patients whose data haven’t been analyzed.

In the discussion section, I would present less details of the studies that you mention and add/comment on some of the above.

Thank you.

Reviewer #2: This study presents the differences in foveal morphology between diabetic and nondiabetic eyes derived from optical coherence tomography angiography and attributed to different race and ethnicity of the patients. The authors managed to provide a well and thoroughly structured study, adding to the literature interesting information and insight into retinal microvasculature. The statistical analysis also seems to be appropriately conducted. The following are a few points that need to be taken into account.

Major points

Lines 140-142: It should be mentioned whether any severe NPDR or PDR cases occurred during the recruitment of diabetic patients. Were these cases excluded from the study analysis and why?

Lines 161-162: It is mentioned that “447 eyes from 280 patients were included in the study”, whereas in Table 1 the total number of included eyes and patients is n=432 and n=271, respectively. The authors should clarify the exact number of included subjects.

Furthermore, the authors had better revise the provided results in table 1 regarding “Years Since Diabetes Diagnosis”. More specifically, they should add whether the data are presented as mean (SD) and also explain what the subcategory “missing” exactly indicates.

Lines 193-202: Whether the provided data arose after adjusting for age, gender, hypertension and HbA1c should additionally be reported, since there is not a corresponding table to clarify this point.

Lines 217: “..while NH black patients had the largest FAZ area.”: It would be helpful, if the authors reported the group to which this outcome refers, i.e. the mild to moderate NPDR group, since among non-diabetic subjects, according to table 3, Hispanic and Asian patients had larger FAZ area than both NH white and NH black people.

Lines 51-52, 217-218, 262-263, 308-309: Only a few of the reported differences have been found to be statistically significant. More specifically, racial and ethnic differences in FAZ area among patients with diabetes without retinopathy did not show statistical significance. The difference in FAZ area between NH black and NH white participants at baseline was not statistically significant either. These points should be rendered clear while presenting the results. The conclusions of the study should also be drawn appropriately based on the data presented.

Lines 237-238: The authors should explain in more detail the mentioned correlation and agreement of their findings with those of Poon et al., regarding both RNFL thickness and MDB thickness and area, in order to make this correlation sound and clear enough for the reader, since different retinal parameters are investigated in the studies.

Lines 270-271 (Figure 1): It is advisable that the authors add the magnification of the pictures. The magnification should be similar, so that the pictures are comparable.

Minor points

Lines 60-61: It is advisable that “retinal microvasculature” are included as keywords.

Lines 127-130: Some of the parameters were automatically calculated. The authors should also explain how the other OCTA parameters were calculated or add the relating reference.

Lines 134-136: The authors should make a reference in the methods section of the manuscript to the final four race/ethnicity groups that are used in the study analyses.

Line 138: The explanation of the abbreviation "MAR" in logMAR should be included, since it has not been mentioned previously in the manuscript.

I would like to look at a revised version of the manuscript.

6. PLOS authors have the option to publish the peer review history of their article (what does this mean?). If published, this will include your full peer review and any attached files.

Reviewer #1: No

Reviewer #2: No

---

## [Author Response · Author response to Decision Letter 0]

18 Sep 2021

Please note that we have provided the responses to reviewers in a side-by-side table format for ease of review. It is a separate attachment to this submission.

---

## [Editor Report · Decision Letter 1]

7 Oct 2021

Racial and ethnic differences in foveal avascular zone in diabetic and nondiabetic eyes revealed by optical coherence tomography angiography

PONE-D-21-20816R1

Dear Dr. Stewart,

We’re pleased to inform you that your manuscript has been judged scientifically suitable for publication and will be formally accepted for publication once it meets all outstanding technical requirements.

Kind regards,

Demetrios G. Vavvas

Academic Editor

PLOS ONE
---

## [Editor Report · Acceptance letter]

13 Oct 2021

PONE-D-21-20816R1 

Racial and ethnic differences in foveal avascular zone in diabetic and nondiabetic eyes revealed by optical coherence tomography angiography 

Dear Dr. Stewart:

I'm pleased to inform you that your manuscript has been deemed suitable for publication in PLOS ONE. Congratulations! Your manuscript is now with our production department. 

Kind regards, 

on behalf of

Prof. Demetrios G. Vavvas 

Academic Editor

PLOS ONE